# Flexible Option Learning

**Martin Klissarov**
Mila, McGill University
`martin.klissarov@mail.mcgill.ca`

**Doina Precup**
Mila, McGill University and DeepMind
`dprecup@cs.mcgill.ca`

## Abstract

Temporal abstraction in reinforcement learning (RL), offers the promise of improving generalization and knowledge transfer in complex environments, by propagating information more efficiently over time. Although option learning was initially formulated in a way that allows updating many options simultaneously, using off-policy, intra-option learning (Sutton, Precup & Singh, 1999), many of the recent hierarchical reinforcement learning approaches only update a single option at a time: the option currently executing. We revisit and extend intra-option learning in the context of deep reinforcement learning, in order to enable updating all options consistent with current primitive action choices, without introducing any additional estimates. Our method can therefore be naturally adopted in most hierarchical RL frameworks. When we combine our approach with the option-critic algorithm for option discovery, we obtain significant improvements in performance and data-efficiency across a wide variety of domains.

## 1 Introduction

Temporal abstraction is a fundamental component of intelligent agents as it allows for explicit reasoning at different timescales. The options framework [Sutton et al., 1999b] provides a clear formalism for such abstractions and proposes efficient ways to learn directly from the environment's reward signal. As the agent needs to learn about the value of a possibly large number of options, it is crucial to maximize each interaction to ensure sample efficiency. Sutton et al. [1999b] therefore propose the intra-option value learning algorithm, which updates all *consistent* options simultaneously from a single transition.

Recently there have been important developments on how to learn or discover options from scratch when using function approximation [Bacon et al., 2016, Smith et al., 2018, Riemer et al., 2018, Bagaria and Konidaris, 2019, Zhang and Whiteson, 2019, Harutyunyan et al., 2017]. However, most of recent research only updates a single option at a time, that is, the option selected by the agent in the sampled state. This is perhaps due to the fact that consistency between options is less likely to arise naturally when using function approximation.

We propose a scalable method to better exploit the agent's experience by updating all relevant options, where relevance is defined as the likelihood of the option being selected. We present a decomposition of the state-option distribution which we leverage to remain consistent in the function approximation setting while providing performance improvements similarly to the ones shown in the tabular setting [Sutton et al., 1999b].

When an option set is given to the agent and only the value of each option remains to be learned, updating all relevant options lets the agent determine faster when to apply each of them. In the case where all options components are learned from scratch, our approach can additionally help mitigate the issue of degenerate solutions [Harb et al., 2017]. Indeed, such solutions are likely due to the "rich-get-richer" phenomenon where an updated option has more chance of being picked again

35th Conference on Neural Information Processing Systems (NeurIPS 2021).

when compared to a randomly initialized option. By avoiding degenerate solutions, one can obtain temporally extended options which leads to meaningful and interpretable behavior.

Unlike recent approaches that leverage inference to update all options [Daniel et al., 2016, Smith et al., 2018, Wulfmeier et al., 2020], our method is naturally compatible with many state of the art policy optimization frameworks [Mnih et al., 2016, Schulman et al., 2017, Fujimoto et al., 2018] and requires no additional estimators. We empirically verify the merits of our approach on a wide variety of domains, ranging from gridworlds using tabular representations [Sutton et al., 1999b, Bacon et al., 2016], control with linear function approximation [Moore, 1991], continuous control [Todorov et al., 2012, Brockman et al., 2016] and vision-based navigation [Chevalier-Boisvert, 2018].

## 2 Related Work

The discovery of meaningful temporally extended behaviour has previously been explored through many different avenues. One such avenue is by learning to reach goals in the environment . This can be done in various ways such as identifying bottleneck states [McGovern and Barto, 2001, Şimşek et al., 2005, Menache et al., 2002, Machado et al., 2017] or through feudal RL [Dayan and Hinton, 1993, Vezhnevets et al., 2017, Nachum et al., 2018].

Another avenue of research is through the options framework [Sutton et al., 1999b] where the discovery of temporal abstractions can be done directly through the environment's reward signal. This framework has lead to recent approaches [Levy and Shimkin, 2011, Bacon et al., 2016, Harb et al., 2017, Riemer et al., 2018, Tiwari and Thomas, 2018] that rely on policy gradient methods to learn the option policies. As policy gradient methods are derived for the on-policy case, these approaches have only updated the sampled option within a state. Our work complements them by providing a way to transform single-option updates into updates that consider all relevant options without introducing any additional estimators.

Within the options framework, Harutyunyan et al. [2017] have recently decoupled the way options terminate from the target termination functions. This allowed for learning longer options as if their duration was less extended in time. This can be beneficial as shorter options are shown to produce better solutions and longer options are preferred for planning [Mann et al., 2015]. By contrast, our approach looks at updating all relevant options through the experience generated by any such option, irrespective of the duration.

Another line of work is proposed by Daniel et al. [2016] in which options are considered latent unobserved variables which allows for updating all options. The authors adopt an expectation-maximization approach which assumes a linear structure of the option policies. Smith et al. [2018] alleviate this assumption and derive a policy gradient objective that improves the data-efficiency and interpretability of the learned options. A conceptually related work is proposed by Wulfmeier et al. [2020] which leverages dynamic programming to infer option and action probabilities in hindsight. As these approaches rely on inference methods to perform updates, it is not directly compatible with other option learning methods nor with various policy optimization algorithms, which is another important feature of our approach.

## 3 Background and Notation

We assume a Markov Decision Process $\mathcal{M}$, defined as a tuple $\langle \mathcal{S}, \mathcal{A}, r, P \rangle$ with a finite state space $\mathcal{S}$, a finite action space $\mathcal{A}$, a scalar reward function $r : \mathcal{S} \times \mathcal{A} \rightarrow Dist(\mathbb{R})$ and a transition probability distribution $P : \mathcal{S} \times \mathcal{A} \rightarrow Dist(\mathcal{S})$. A policy $\pi : \mathcal{S} \rightarrow Dist(\mathcal{A})$ specifies the agent's behaviour, and the value function is the expected discounted return obtained by following $\pi$ starting from any state: $V_\pi(s) \doteq \mathbb{E}_\pi \left[ \sum_{i=t}^\infty \gamma^{t-i} r(S_i, A_i) \big| S_t = s \right]$, where $\gamma \in [0, 1)$ is the discount factor. The policy gradient theorem [Sutton et al., 1999a] provides the gradient of the expected discounted return from an initial state distribution $d(s_0)$ with respect to a parameterized stochastic policy $\pi_\zeta(\cdot|s)$: $\frac{\partial J(\zeta)}{\partial \zeta} = \sum_s d^\gamma(s) \sum_a \frac{\partial \pi_\zeta(a|s)}{\partial \zeta} Q_\pi(s, a)$ where $d^\gamma(s) = \sum_{s_0} d(s_0) \sum_{t=0}^\infty \gamma^t P^\pi(S_t = s | S_0 = s_0)$ is the discounted state occupancy measure.

**Options.** The options framework [Sutton et al., 1999b] provides a formalism for abstractions over time. An option within the option set $\mathcal{O}$ is composed of an intra-option policy $\pi(a|s, o)$ for selecting actions, a termination function $\beta(s, o)$ to determine how long an option executes and an initiation set

$I(s, o)$ to restrict the states in which an option may be selected. It is then possible to choose which option to execute through the policy over options $\mu : \mathcal{S} \to Dist(\mathcal{O})$. The set of options along with the policy over options defines an option-value function which can be written recursively as,

$$Q_{\mathcal{O}}(s, o) \doteq \sum_a \pi(a|s, o) r(s, a) + \gamma \sum_{s', o'} P_{\pi, \mu, \beta}(s', o'|s, o) Q_{\mathcal{O}}(s', o')$$

where $P_{\pi, \mu, \beta}(s', o'|s, o)$ is the probability of transitioning from $(s, o)$ to any other state-option pair in one step in the MDP. This probability is defined as,

$$P_{\pi, \mu, \beta}(s', o'|s, o) \doteq p_{\mu, \beta}(o'|s', o) \sum_a P(s'|s, a) \pi(a|s, o)$$

where $p_{\mu, \beta}(o'|o, s') = \beta(s', o) \mu(o'|s') + (1 - \beta(s', o)) \delta_{o, o'}$ is the upon-arrival option probability and $\delta$ is the Kronecker delta. Intuitively, this distribution considers whether or not the previous option terminates in $s'$, and accordingly either chooses an option or continues with the previous one.

## 4  Learning Multiple Options Simultaneously

In this work we build upon the *call-and-return* option execution model. Starting from a state $s$, the agent picks an option according to the policy over options $o \sim \mu(\cdot|s)$. This option then dictates the action selection process through the intra-option policy $a \sim \pi(\cdot|s, o)$. Upon arrival at the next state $s'$, the agent queries the option's termination function $\beta(s', o)$ to decide whether the option continues or a new option is to be picked. This process then continues until the episode terminates.

This execution model, coupled with the assumption that the options are Markov, will let us leverage a notion of how likely an option is to be selected, irrespective of the actual sampled option. This measure will then be used to weight the updates of each possible option.

### 4.1  Option Evaluation

In this section we present an algorithm that updates simultaneously the option value functions of all the options, written as $Q_{\mathcal{O}}(s, o; \theta)$ and parameterized by $\theta$. Let's recall the on-policy one-step TD update for state-option pairs,

$$\theta_{t+1} = \theta_t + \alpha \big( R_{t+1} + \gamma \hat{Q}(S_{t+1}, O_{t+1}) - \hat{Q}(S_t, O_t) \big) \nabla_{\theta_t} \hat{Q}(S_t, O_t)$$

$$\theta_{t+1} = \theta_t + \alpha \big( R_{t+1} + \gamma \theta_t^\top \phi(S_{t+1}, O_{t+1}) - \theta_t^\top \phi(S_t, O_t) \big) \phi(S_t, O_t)$$

$$= \theta_t + \alpha \big( \underbrace{R_{t+1} \phi(S_t, O_t)}_{\mathbf{b}_t} + \underbrace{\phi(S_t, O_t)(\phi(S_t, O_t) - \gamma \phi(S_{t+1}, O_{t+1}))^\top}_{\mathbf{A}_t} \theta_t \big)$$

$$\theta_{t+1} = \theta_t + \alpha(\mathbf{b}_t - \mathbf{A}_t \theta_t) \tag{1}$$

where the option value functions are linear functions, i.e. $Q_{\mathcal{O}}(s, o; \theta) \doteq \theta_t^\top \phi(s, o)$ of the parameters $\theta_t$ at time $t$ and the features $\phi(s, o)$. Since the random variables within the update rule, summarized through the vector $\mathbf{b}_t$ and the matrix $\mathbf{A}_t$, are sampled from interacting with the environment, the stability and convergence properties can be verified by inspecting its expected behaviour in the limit, written as,

$$\theta_{t+1} = \theta_t + \alpha(\mathbf{b} - \mathbf{A}\theta_t)$$

where $\mathbf{A} = \lim_{t \to \infty} \mathbb{E}[\mathbf{A}_t]$ and $\mathbf{b} = \lim_{t \to \infty} \mathbb{E}[\mathbf{b}_t]$. As the matrix $\mathbf{A}$ has a multiplicative effect on the parameters, it will be crucial for this matrix to be positive-definite in order to assure the stability of the algorithm, and subsequently its convergence. In the on-policy case, this matrix is written as:

$$\mathbf{A} = \sum_{s, o} \lim_{t \to \infty} P_{\pi, \mu, \beta}(S_t = s, O_t = o) \phi(s, o) \bigg( \phi(s, o) - \gamma \sum_{s', o'} P_{\pi, \mu, \beta}(s', o'|s, o) \phi(s', o') \bigg)^\top$$

where $P_{\pi, \mu, \beta}(S_t = s, O_t = o)$ is the probability of being in a particular state-option pair at time $t$. In order to update all options, we will exploit the form of this state-option distribution to preserve the positive definite property of the matrix $\mathbf{A}$. In the following lemma we first show that this limiting distribution exists under some assumptions and we derive a useful decomposition. To do so, we

introduce the following two assumptions. Regarding Assumption 1, it is common in standard RL to assume that the policy induces ergodicity on the state space, which we extend to all option policies. Assumption 2 is specific to the hierarchical setting and is a way to ensure irreducibility, that is, that each state-option pair is reachable. It would be possible to find less strict assumptions by having access to more information about how the MDP is constructed.

**Assumption 1.** *The option policies $\{\pi(a|s, o) : o \in \mathcal{O}\}$ each induce an ergodic Markov chain on the state space $\mathcal{S}$.*

**Assumption 2.** *The termination function $\beta(s, o)$ and policy over options $\mu(o|s)$ have strictly positive probabilities in all state-option pairs.*

**Lemma 1.** *Under assumptions 1-2 the following limit exists,*

$$\lim_{t \to \infty} P_{\pi,\mu,\beta}(S_t = s, O_t = o | S_0, O_0) = d_{\pi,\mu,\beta}(s, o) \tag{2}$$

*and the augmented chain process is an ergodic Markov chain in the augmented state-option space. Moreover the stationary distribution can be decomposed in the following form,*

$$d_{\pi,\mu,\beta}(s, o) = \sum_{\bar{o}} \bar{d}_{\pi,\mu,\beta}(\bar{o}, s) p_{\mu,\beta}(o|s, \bar{o}) \tag{3}$$

*where $p_{\mu,\beta}(o|s, \bar{o})$ is the probability over options upon arriving in a state.*

**Remarks.** In the derivation we have avoided mentioning the concept of initiation sets such that we would not encumber the notation and presentation. The derivation would be identical if we had assumed that the initation set for each option is equal to the state space, which has been done for most of recent work on options [Bacon et al., 2016]. Another way to reconcile our method with the original options framework would be to consider interest functions [Khetarpal et al., 2020], which are a generalization of the concept of initiation sets. We could then add an assumption similar to Assumption 2 on the interest function by only allowing positive values. This would let us heavily favor a subset of options for a certain region of the state space. However, this assumption would still preclude us from using hard constraints on the option set, such as option affordances [Khetarpal et al., 2021]. More empirical evidence would be necessary to investigate whether this is a limitation in practice.

Ideally, if we seek to update all options in a state, we could weight each of these updates by the stationary distribution $d_{\pi,\mu,\beta}(s, o)$. However, in most case, it is impractical to assume access to this distribution. A possible solution would be to estimate this quantity by making use of recent approaches [Hallak and Mannor, 2017, Xie et al., 2019, Zhang et al., 2020]. Instead, we propose to leverage the decomposition presented in Equation 3.

Indeed, this decomposition highlights the dependency on the upon-arrival option distribution, $p_{\mu,\beta}(o|s, \bar{o})$, which is a known quantity as it is only dependent of the agent's policy over options $\mu(o|s)$ and termination function $\beta(s, \bar{o})$. Although $\bar{d}_{\pi,\mu,\beta}(s, \bar{o})$ is itself unknown, we can collect its samples by adopting the call-and-return option execution model. More concretely, under this model, an option which is previously selected leads us to a state of interest at time $t$. These random variables, being sampled on-policy, follow the distribution $\bar{d}_{\pi,\mu,\beta}(s, \bar{o})$. Upon arriving in any state of interest $S_t$ with option $O_{t-1}$, although an option $O_t$ will be sampled according to call-and-return, we can use the distribution $p_{\mu,\beta}(O_t = o|S_t, O_{t-1})$ to estimate how likely it would have been for each option $o$ to be selected. The update rule would become,

$$\theta_{t+1} = \theta_t + \alpha \sum_o p_{\mu,\beta}(O_t = o|S_t, O_{t-1}) \big( \mathbb{E}[U_t|S_t, O_t = o] - \theta_t^\top \phi(S_t, O_t = o) \big) \phi(S_t, O_t = o)$$

$$\tag{4}$$

where $\mathbb{E}[U_t|S_t = s, O_t = o]$ is the target for a given state and option pair. In Section 4.3 we show how to obtain this quantity without using any additional estimation or simulation-resetting. Our update rule would then preserve the positive-definite property of the matrix $\mathbf{A}$.

## 4.2 Option Control

In this section we extend the approach from evaluating the option value functions to the case where we seek to learn all option policies simultaneously. To do so we build upon the option-critic (OC)

update rules derived by Bacon et al. [2016]. Let us first recall the intra-option policy gradient,

$$\sum_{s,o} d^\gamma_{\pi,\mu,\beta}(s,o) \sum_a \frac{\partial \pi_\zeta(a|s,o)}{\partial \zeta} Q_\mathcal{O}(s,o,a) \tag{5}$$

where, $d^\gamma_{\pi,\mu,\beta}(s,o) = \sum_t \gamma^t P_{\pi,\mu,\beta}(S_t = s, O_t = o)$, is the $\gamma$-discounted occupancy measure over state-option pairs. As in the previous section, we wish to adopt the same call-and-return option execution model, in particular as it is also used by the option-critic algorithm in order to perform stochastic gradient updates following Equation 5. When updating all options within a state $s$, we notice that we can leverage the structure of distribution state-option occupancy measure,

$$d^\gamma_{\pi,\mu,\beta}(s,o) = \sum_{\bar{o}} p_{\mu,\beta}(o|s,\bar{o}) \sum_{\bar{s}} p_\pi(s|\bar{s}) d^\gamma_{\pi,\mu,\beta}(\bar{s},\bar{o}) \tag{6}$$

By expanding the state-occupancy measure we can separate unknown probabilities, that is the next state distribution $p_\pi(s|\bar{s})$ and the previous state-option pair $d^\gamma_{\pi,\mu,\beta}(\bar{s},\bar{o})$, from the known upon-arrival option distribution, that is $p_{\mu,\beta}(o|s,\bar{o})$. Executing Markov options in a call-and-return fashion will then provide us samples from the unknown distributions, and for each sampled $(S_t, O_{t-1})$-pair we can weight the likelihood of choosing option $o$ at time $t$. Concretely, this translates into the following update rule,

$$\zeta_{t+1} = \zeta_t + \alpha_\zeta \sum_o p_{\mu,\beta}(O_t = o|S_t, O_{t-1}) \frac{\log \partial \pi_\zeta(A_t|S_t, O_t = o)}{\partial \zeta_t} Q_\mathcal{O}(S_t, O_t = o, A_t) \tag{7}$$

We highlight that the update rules for option control do not rely on Assumption 1 and 2, as these were specific to the evaluation setting. By updating all options within a state, we can expect better sample efficiency and final performance. Moreover, this may also contribute to reducing variance. Indeed, the update rule in Equation 7 can be seen as a variation on the all-action policy gradient [Sutton et al., 1983, Petit et al., 2019], which updates all actions applied to the distribution of options. Such all-action or all-option updates can be derived as applications of the extended [Bratley et al., 1987] form of conditional Monte-Carlo estimators [Hammersley, 1956, Liu et al., 2019], which provide a conditioning effect on the updates that tend to reduce the variance. As we will see in our experiments, such variance reduction can be observed, especially with tabular and linear function estimators.

It is possible to combine evaluation and control into a single method. We do so by instantiating these update rules within the option-critic (OC) algorithm which we present in Algorithm 1 in appendix A.1 and name it Multi-updates Option Critic (MOC) algorithm.

### 4.3 Suitable targets

We have proposed a way in which all relevant options can be updated simultaneously. In the case of option evaluation we relied on having access to appropriate targets $\mathbb{E}[U_t|S_t = s, O_t = o]$ for each option within a state. However, as we execute options in the call-and-return model and we do not assume access to the simulator for arbitrary resetting, we only have one sampled action for each state: the one taken by the executing option. To leverage this experience for any arbitrary option, we will leverage standard importance sampling [Precup et al., 2000]. Using importance sampling naturally excludes the possibility that options be defined through deterministic policies, which can be seen as a limitation of our approach. That being said, importance sampling is not necessary for the option control algorithm.

The importance sampling target for any arbitrary option $\tilde{o}$ given the action $a$ in state $s$ generated by the executing option $o$ is the following,

$$U_t^\rho(s,\tilde{o}) = \frac{\pi(A_t = a|S_t = s, O_t = \tilde{o})}{\pi(A_t = a|S_t = s, O_t = o)} R_{t+1} + \frac{p_{\mu,\beta}(O_{t+1} = o'|S_{t+1} = s', O_t = \tilde{o})}{p_{\mu,\beta}(O_{t+1} = o'|S_{t+1} = s', O_t = o)} \gamma \theta_t^\top \phi(s', o')$$

It is straightforward to show that this target is unbiased and corrects for the off-policyness introduced by updating any other option than the sampled one, leading to consistency [Sutton et al., 1999b]. It would be possible to consider different approaches to correct for difference between intra-option policies [Mahmood et al., 2014, Li et al., 2015, Harutyunyan et al., 2016, Jiang and Li, 2016, Jain and Precup, 2018] which we leave for future work.

### 4.4 Controlling the number of updates

A possible drawback of updating all options is the prospect of reducing the degree of diversity in the option set. A possible remedy would be to favor diversity, or similar measures, through reward shaping [Gregor et al., 2016, Eysenbach et al., 2018, Pathak et al., 2019, Klissarov and Precup, 2020]. In this work, we instead introduce a hyperparameter $\eta$ which represents the probability of updating all options, as opposed to updating only the sampled option. In our experiments we will notice that high values of $\eta$ are not necessarily detrimental to the expressivity of the option set.

Another interesting property of this hyperparameter, which we do not leverage in this work, is that it could be use in a state dependent way (i.e. $\eta(s)$) without changing the stability of the algorithm. This generalization adds flexibility to how we want to update options. In particular, it could be a way to limit the variance of the algorithm by only performing such off-policy updates for state-option pairs where the importance sampling is below a threshold.

## 5 Experiments

We now validate our method on a wide range of tasks and try to assess the following questions: (1) By updating all relevant options are we able to improve the sample-efficiency and final performance? (2) Does our approach induce temporally extended options without the need of regularization? (3) Is the learned option set diverse and useful?

Most of our experiments are done in a transfer learning setting. Our method is compatible with any modern policy optimization framework and we validate this by building our hierarchichal implementations on top of the synchronous variant of the asynchronous advantage actor-critic algorithm [Mnih et al., 2016], that is the A2C algorithm, as well as the proximal policy optimization algorithm (PPO) [Schulman et al., 2017]. All hyperparameters are available in the appendix. Importantly, for each set of experiments we use the same value of $\eta$. For all experiments, the shaded area represents the $80\%$ confidence interval.

We would also like to emphasize that although our implementation is based upon the option-critic algorithm [Bacon et al., 2016], our approach can be straightforwardly adapted to most hierarchical RL algorithms [Riemer et al., 2018, Jain et al., 2018, Jinnai et al., 2019, Zhang and Whiteson, 2019, Bagaria and Konidaris, 2019].

### 5.1 Tabular Domain

*Experimental Setup:* We first evaluate our approach in the classic FourRooms domain [Sutton et al., 1999b] where the agent starts in a random location and has to navigate to the goal in the right hallway. As we are interested in the algorithms' sample efficiency, we plot the number of steps taken per episodes.

In Figure 1a, we work under the assumption that the options have been given and remain fixed throughout learning. The only element being updated is the option value functions, $Q_{\mathcal{O}}(s, o)$, which is used by the policy over options to select among options. We compare our approach, leveraging the multi-options update rule in Equation 4, to the case of only updating the sampled option within a state. The options used are the 12 hallway options designed in Sutton et al. [1999b] (see their Figure 2 for a brief summary). In this setting, our approach achieves significantly better sample efficiency, illustrating how our method can accelerate learning the utility of a set of well-suited options.

We now move to the case where all the options' parameters are learned and leverage the learning rule for control in Equation 7. In this setting we compare our approach, multi-updates option critic (MOC), to the option-critic (OC) algorithm as well as a flat agent, denoted as actor-critic (AC). Both hierarchical agents outperform the flat baselines, however, our agent is able to reach the same performance as OC in about half the number of episodes. Moreover, we notice that MOC shows less variance across runs, which is in accordance with previous work investigating the variance reduction effect of expected updates [Sutton et al., 1983, Petit et al., 2019].

In Figure 6 in the appendix, we verify the kind of options that are being learned under both the option-critic and our approach. For each option we plot the best action in each state and we highlight in yellow whether the option is likely to be selected by the policy over options. We first notice that for the OC agent, there is an option that is selected much more often, whereas the MOC agent learns

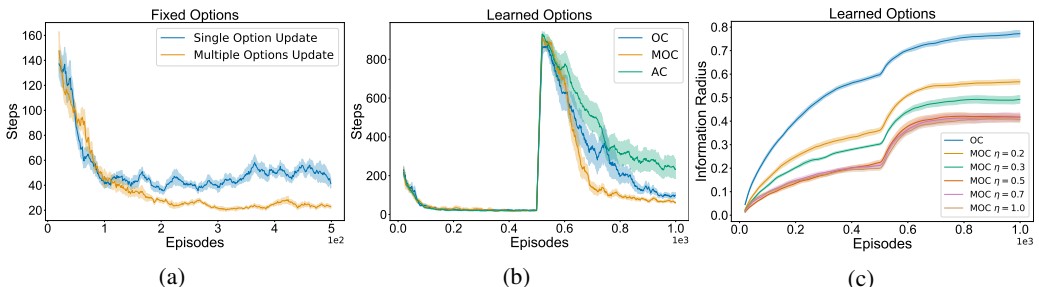

(a)                             (b)                             (c)

Figure 1: **FourRooms domain.** In a) we used the fixed set of hallway options as defined in Sutton et al. [1999b] and only learned the option-value functions. In b) all the parameters are learned from scratch before the goal location is change mid-training. In c) we plot the information radius between options as a measure of diversity. Results are averaged over 50 independent runs.

a good separation of the state space. Moreover, we notice in purple states (where the option is less likely to be chosen), the OC agent's actions do not seem to lead to a meaningful behaviour in the environment, which is in contrast to our approach. As a potential benefit, by sharing information between options, the MOC agent will tend to be more robust to the environment's stochasticity as well as to the learning process of the policy over options.

Finally, we also would like to point out a possible challenge in our approach that we have mentioned in Section 4. It is possible that by learning all options simultaneously we may reduce the diversity between options. We verify this in Figure 1c where we plot the information radius [Sibson, 1969] between options as a measure of diversity. We plot the OC agent as well as our approach, MOC, with different values of the $\eta$ hyperparameter which controls the degree to which we apply updates to all options. We notice indeed a clear different between the OC agent and our agent when $\eta = 1.0$ (only updates to all options). However, intermediate values of $\eta$ provide an effective to increase the information radius. As in this experiment we use tabular representations, the difference between updated options and non-updated options is accentuated. In the function approximation case we will see that the concern about the diversity of options is mitigated.

## 5.2 Linear Function Approximation

*Experimental Setup:* We now perform experiments in the case where the value function is estimated through linear function approximation. The environment we consider is a sparse variant of the classic Mountain Car domain [Moore, 1991, Sutton and Barto, 2018] where a positive reward is given upon reaching the goal. We consider the transfer setting where after half the total number of timesteps the gravity is doubled.

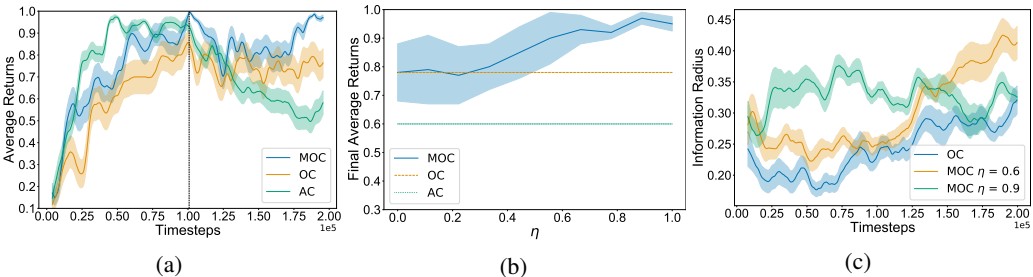

(a)                             (b)                             (c)

Figure 2: **MountainCar domain.** In a) we plot the performance of our agent, MOC, the option-critic and a flat baseline. The options are learned from scratch. At mid-training, we modify the gravity to double its original value. In b) we perform a hyperparamter study on the value of $\eta$ which controls the amount of updates to all options. In c) we plot the information radius between options as a measure of expressivity and notice that higher values of $\eta$ do not affect it negatively. The performance is averaged across 20 independent runs.

In Figure 2a, we observe similar results to the tabular case, that is, hierarchical agents perform better than standard RL in the transfer task. MOC also improves upon OC's performance and sample efficiency. We verify how this improvement in performance translates with respect to the value of $\eta$, the hyperparameter controlling for the probability of updating all options. In Figure 2b we can observe an almost monotonic relationship between $\eta$ and the algorithm's final performance, which confirms the effectiveness of our approach when using function approximation.

Finally, in Figure 2c we verify the information radius between options for different values of $\eta$. We notice that unlike the tabular case where higher values of $\eta$ lead to a smaller information radius, there is no clear correlation in this case. Perhaps surprisingly we notice that in the first task high values of $\eta$ lead to an increased information radius. This might indicate that at the beginning of training, the agent can learn to specialize faster by updating all options.

## 5.3 Experiments at Scale

### 5.3.1 Continuous Control

*Experimental Setup:* We turn our attention to the deep RL setting where all option components are estimated through neural networks. We first perform experiments in the MuJoco domain [Todorov et al., 2012] where both states and actions are continuous variables. We build on classic tasks [Brockman et al., 2016] and modify them to make the learning process more challenging. The description of each task as well as a visual representation is available in the appendix.

We leverage the PPO algorithm [Schulman et al., 2017] to build our hierarchical agents. We also compare our approach, MOC, to the interest option-critic (IOC) [Khetarpal et al., 2020] which has been shown to outperform the option-critic algorithm derived for the continuous control setting [Harb et al., 2017, Klissarov et al., 2017]. Additionally, we compare to a flat agent using PPO and to the inferred option policy gradient (IOPG) algorithm [Smith et al., 2018], an HRL algorithm also designed for updating all options by leveraging inference.

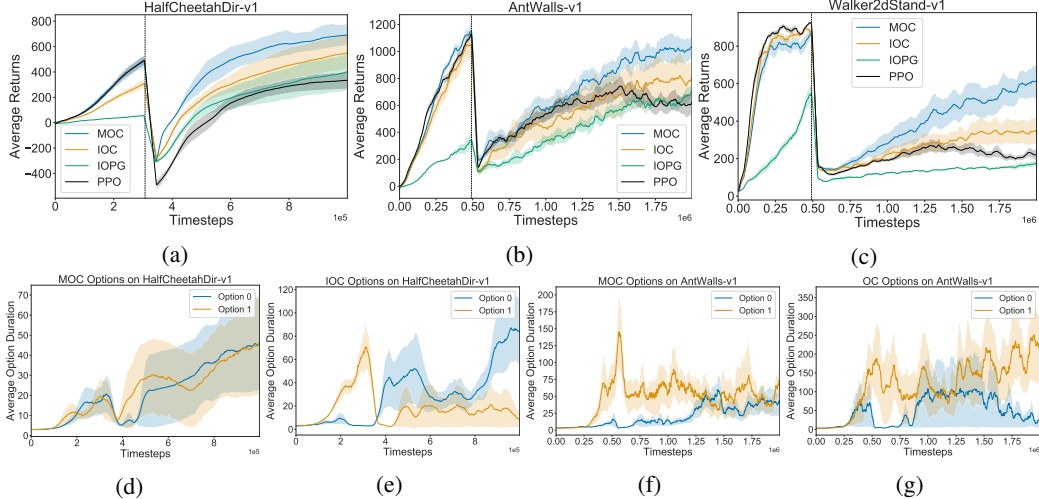

Figure 3: **Transfer on MuJoco.** In a-b-c) we compare the performance of our approach, MOC, to various hierarchical baselines and to a flat agent. At about a quarter into the training process, the original task is modified to a more challenging, but related, setting where the agents must transfer their knowledge. In d-e-f-g), we plot the option duration for both the MOC agent and the OC baseline. We notice that our approach tends to produce options that are extended in time, which can lead to meaningful behavior. All experiments are plotted using 10 independent runs.

Across environments in Figure 3abc we witness that MOC shows better sample efficient and performance compared to the other hierarchical agents as well as the PPO baseline. In particular, although IOPG updates all options similarly to our approach, this doesn't translate in a practical advantage. This highlights the adaptability of the update rules we propose to any state-of-the-art policy optimization algorithm. We also emphasize that in all these experiments, the same value of $\eta$ is used, making it a general approach.

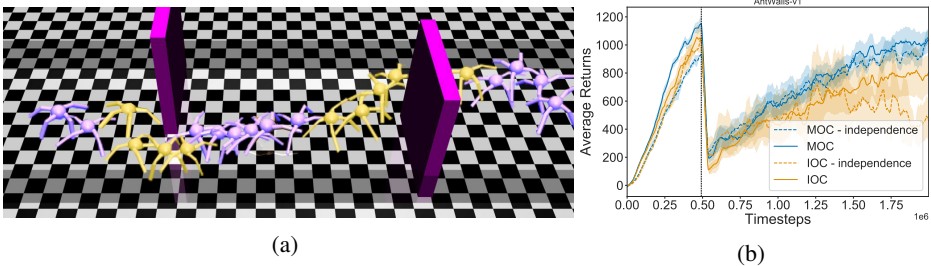

(a)

(b)

Figure 4: a) **Option visualization.** In this experiment, AntWalls-v1, the agent has first learned how to walk in the forward direction before we add obstacles (walls) which it needs to circumvent. We highlight in purple the states in which our agent uses the first option and in yellow the second option. b) Performance of MOC and IOC algorithms for different neural network architectures. Independence indicates that the second last hidden layer does is not shared across optiosn.

In Figure 3defg we investigate the options' duration for both the IOC agent and our approach. We investigate both methods using two options, so as to provide more interpretable results We argue that such a metric can be an important indicator of the usefulness of options, which are meant to be temporally extended. We notice that both in HalfCheetahDir-v1 and AntWalls-v1, both options learned by MOC are used by the agent over multiple timesteps. This is in contrast to the IOC agent where options at the end of training exhibit an imbalance. We also highlight that although MOC improves this aspect of option learning, the shaded are is still large which indicates that it does not fully solve the problem.

For the MOC agent, it is interesting to notice temporally extended options are truly present once the agent faces a new task. This perhaps suggests that future inquiry should look into whether challenging the agent with multiple tasks (such as in the continual learning setting) is actually a catalyst for the discovery of options.

In Figure 4a, we provide a visualization of meaningful behaviour exhibited by the MOC agent. We notice that in the AntWalls-v1 environment, one agent learns an option to move forward (highlighted in purple), while the other one is used to move around the walls (highlighted in yellow). We provide additional visualization (including from the IOC agent) in the appendix. We mention that although such meaningful behaviour was witnessed more often in our approach, we did not completely avoid degenerate solutions for all random seeds or all values of the learning rate.

In our experiments we compared HRL algorithms when using two options. We now verify how MOC and IOC scale when we grow the option set to four and eight options. In Table 1, we notice that most of the gains from updating all options are persistent for each cardinality. However, we notice eventually a significant drop in performance when using 8 options. This could attributed to the fact that the domains we experiment with are simple enough that such a high

Table 1: Walker2dStand-v1

| Algorithm | 2 options | 4 options | 8 options |
|---|---|---|---|
| MOC | 617 (98) | 574 (76) | 535 (81) |
| IOC | 326 (92) | 238 (55) | 212 (64) |

number of options is unnecessary and leads to reduced sample efficiency. One way to verify this would be to apply the same algorithm in a continual learning setting. It could also be the case that the OC baseline on which we implement our flexible update rules could be itself improved for better performance. More experiments are needed to investigate these questions, which we leave for future work.

Finally, a promising possibility of our method is to re-organize the way parameters are shared between options. The neural network architecture used by OC-based algorithms (including ours) shares most of the parameters between options: only each head (last layer of the network) is specific to each option and independent of others. In Figure 4b, we investigate the performance of MOC and IOC when the parameters of the second to last hidden layer are also independent for each option. As we notice through the dashed lines, MOC remains more robust to this configuration. This would indicate

that by using our proposed update rules, more sophisticated network architectures could be devised without sacrificing sample efficiency.

### 5.3.2  MiniWorld Vision Navigation

*Experimental Setup:* In this section we verify how our approach extends to tasks with high dimensional inputs such as the first-person visual navigation domain of MiniWorld [Chevalier-Boisvert, 2018]. We provide a description for each environment in the appendix.

In MiniWorld-YMazeTransfer-v0, the transfer is not drastic and the flat agent is able to recover well in the transfer task. Moreover, MOC seemed to be slower to learn at first, although it was able to transfer with a smaller drop in performance. For the more challenging MiniWorld-WallGapTransfer-v0, we notice that HRL algorithms are more relevant and adapt better to the change, with MOC being more sample efficient.

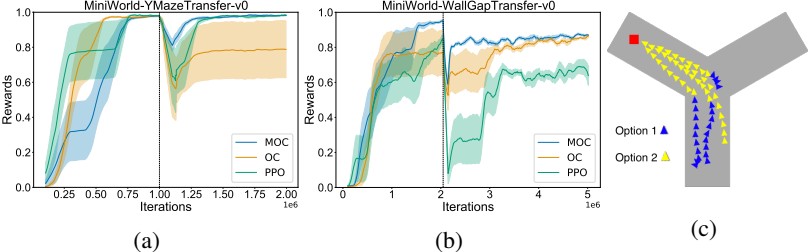

(a)  (b)  (c)

Figure 5: **Transfer in MiniWorld.** In a-b) we plot the performance of our approach and compare it to the option-critic (OC) and a flat baseline (PPO). In particular in b) we can appreciate the need for hierarchy when transferring to a similar, but more challenging task. In c) we present qualitative results on the kind of options that are being learned by our agent. In these experiments we average the performance across six independent runs.

Finally, in 5c, we plot the options learned by our MOC agent in the MiniWorld-YMazeTransfer-v0 environment. The first option learns to get to the middle of the maze and locate the red goal. When the goal has been seen, the second option moves towards it. Learning such a structure can potentially help an agent adapt to a new and related task.

## 6  Conclusion

In this work we have proposed a way to update all relevant options for a visited state. Our method does not introduce any additional estimators and is therefore readily applicable to many state-of-the-art RL, and HRL, algorithms in order to improve their performance and sample-efficiency. Updating all options simultaneously has been an integral part of the intra-option learning proposed in the options framework [Sutton et al., 1999b]. This work therefore aims to extend this flexibility to function approximation and deep RL.

Our proposed update rules can be seen as updates done in expectation with respect to options. As such, they are reminiscent of generalizations of standard RL algorithms, such Expected SARSA [van Seijen et al., 2009] and Expected Policy Gradients [Ciosek and Whiteson, 2018]. Amongst other qualities, such approaches have theoretically been shown to reduce variance. A similar analysis could be performed in our framework to verify what theoretical guarantees can be obtained from updating all relevant options.

We also proposed a flexible way to control between updating all relevant options and updating only the sampled one through the hyperparameter $\eta$. Potential future work could investigate whether a state-dependent $\eta$ function, similarly to interest functions [Mahmood et al., 2015], could be used to highlight important transitions, such as bottlenecks, and update all options only in those states. Finally, learning all options simultaneously could be further leveraged in the case where we learn a very large number of them simultaneously [Precup, 2018, Schaul et al., 2015], before deciding which are useful. Such an investigation would most likely be most promising in a continual learning setting. Updating all relevant options also means that we could implement less parameter sharing between options and discover neural network architectures better suited for HRL.

## Acknowledgments and Disclosure of Funding

The authors would like to thank the National Science and Engineering Research Council of Canada (NSERC) for funding this research; Khimya Khetarpal for early discussions on the project and the anonymous reviewers for providing critical and constructive feedback.

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
