# A Appendix

## A.1 Algorithm

The code is available at `https://github.com/mklissa/MOC`.

---

**Algorithm 1** Multi-updates Option Critic

---

Set $s \longleftarrow s_0$
Choose $o$ at $s$ according to $\mu_z(.|s)$
**repeat**
  Choose $a$ according to $\pi_\zeta(a|s,o)$
  Take action $a$ in $s$, observe $s', r$
  Sample termination from $\beta_\nu(s', o)$
  **if** $o$ terminates in $s'$ **then**
    Sample $o'$ according to $\mu_z(.|s)$
  **else**
    $o' = o$
  **end if**
  Define previous option $\bar{o} = o$
  **1. Evaluation step:**
  **for** each option $\tilde{o}$ in the option set $\mathcal{O}$ **do**
    $\delta \leftarrow \mathbb{E}[U^\rho|s,\tilde{o}] - Q_\theta(s,\tilde{o})$ where $U^\rho$ is an importance sampling weighted target
    $\theta \leftarrow \theta + p_{\mu,\beta}(\tilde{o}|s,\bar{o})\alpha\delta\phi(s,\tilde{o})$
  **end for**
  **2. Improvement step**
  **for** each option $\tilde{o}$ in the option set $\mathcal{O}$ **do**
    $\zeta \leftarrow \zeta + p_{\mu,\beta}(\tilde{o}|s,\bar{o})\alpha_\zeta \frac{\partial \log \pi_\zeta(a|s,o)}{\partial \zeta} Q_\theta(s,o,a)$
  **end for**
  $\nu \leftarrow \nu - \alpha_\nu \frac{\partial \beta_\nu(s',o)}{\partial \nu}(Q_\theta(s',o) - V_\theta(s'))$
  $z \leftarrow z + \alpha_z \beta_\nu(s',o)\frac{\partial \mu_z(o'|s')}{\partial z}Q_\theta(s',o')$
  $s \leftarrow s'$
**until** $s'$ is a terminal state

---

## A.2 Tabular experiments

### A.2.1 Implementation Details

For our experiments of the FourRooms domain we based our implementation on [Bacon et al., 2016] and ran the experiments for 500 episodes that last a maximum of 1000 steps with goal located in the right hallway.

In the first experiment we verify whether learning a fixed set of options can be accelerated by our method. We define this fixed set as the hallway options from Sutton et al. [1999b]. As the policies of these options were deterministic and we use importance sampling, we relax them to stochastic policies where the most likely action is the one leading to a hallway.

In the second experiment, after 500 episodes where the goal is located in the right hallway, we move the goal a random location in the bottom left room and augment the stochasticity of the environment. Both of these modifications are done to verify the agent's ability to adapt to change.

For all experiments and all algorithms we performed a learning rate sweep in the rage of $\{2 \cdot 10^i, 4 \cdot 10^i, 6 \cdot 10^i, 8 \cdot 10^i : i = -1, -2, -3\}$. We used the same learning rate for both intra-option updates as well as the updates on the value function. We did so to avoid introducing a larger hyperparameters search for the hierarchical approaches. For the MOC hyperparameter $\eta$, we looked at values in $\{0.2, 0.3, 0.5, 0.7, 1.0\}$. All values are available in Table 2

| Hyperparameter | Value |
|---|---|
| AC lr | 2e-1 |
| OC lr | 8e-1 |
| MOC lr | 8e-1 |
| $\eta$ | 0.3 |

Table 2: Best hyperparameters for the FourRooms domain.

### A.2.2 Option visualization

In the following figure we present a qualitative representation of the options being learned by our method MOC and compare it to the OC method.

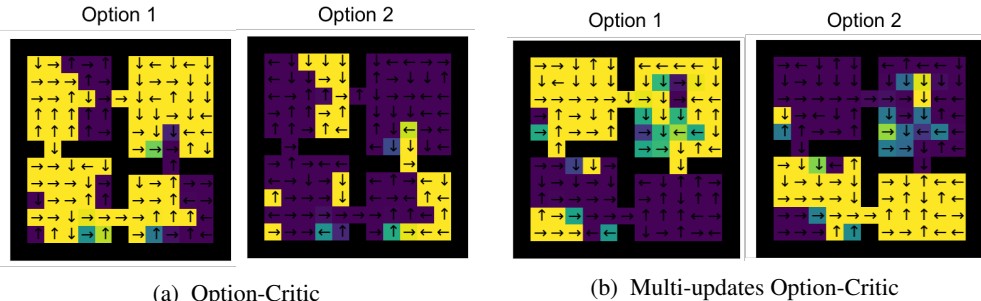

(a) Option-Critic  (b) Multi-updates Option-Critic

Figure 6: **Visualization of the learned options.** For each option, the brightness of the state represent the probability of selecting the said option. For each state and option, we plot by a arrow the most likely action. In a) we present two options learned by the OC algorithm. We notice a certain imbalance between options in the sense that option 1 is almost always taken by the policy over options. In b) we present two options learned by our approach, MOC, and notice a better balance between options. Moreover, even if an option is not likely to be selected, it generally chooses relevant actions, which is not the case for the OC agent.

## A.3   Linear Function Approximation

### A.3.1   Implementation Details

The environment we consider is a sparse variant of the classic Mountain Car domain [Moore, 1991, Sutton and Barto, 2018] where a positive reward is given upon reaching the goal. We consider the transfer setting where after half the total number of timesteps the gravity is doubled.

In this experiment, the critic is a linear function of the features, while the actor is a two layer network with a tanh non-linearity in the hidden layer. The features are defined through four radial basis functions with radii $\{5, 2, 1, 0.5\}$. These RBF kernels are fit by sampling random states from the environment for 100000 steps before the training starts.

For all algorithms we performed a learning rate sweep in the set $\{0.0008, 0.001, 0.003, 0.004, 0.005, 0.006, 0.008, 0.01\}$. As in the tabular case, we used the same learning rate for both intra-option policy updates, the option value function udpates, the termination functions updates as well as the udpates on to the policy over options. We did so to avoid introducing a larger hyperparameters search for the hierarchical approaches. Other hyperparameters values are standard values for the A2C algorithm. For the MOC hyperparameter $\eta$, we did a hyperparameter study for values in $\{0.1, 0.2, 0.3, 0.4, 0.5, 0.7, 0.8, 0.9, 1.0\}$ in order to verify how the algorithm behaves with respect to the number of updates done to all options. The results are plotted in Figure 2b and we see that higher values of $\eta$ lead to generally better final performance. All values are available in Table 3

| Hyperparameter | Value |
|---|---|
| AC lr | 0.005 |
| OC lr | 0.004 |
| MOC lr | 0.004 |
| $\eta$ | 0.9 |
| # steps before update | 5 |
| actor hidden features size | 128 |
| value loss coefficient | 0.5 |
| # RBF kernels | 5 |

Table 3: Best hyperparameters for the MountainCar-v0 domain.

## A.4 Continuous Control

### A.4.1 Tasks Description

We build on classic continuous control domains [Brockman et al., 2016] and modify them to make the learning process more challenging. The first experiment is **HalfCheetahDir-v1**, visualized in Figure 7a , in which the cheetah first learns to walk in the forward direction and then needs to adapt to a change in reward signal which now encourages walking backward. The second experiment is **AntWalls-v1** illustrated in Figure 7b where the agent first learns how to walk forward as in the classic Ant-v0 domain [Brockman et al., 2016]. In the transfer task, we add walls as obstacles and provide a terrain-read which the agent can use to circumvent them. The final experiment is **Walker2dStand-v1** shown in 7c in which the walker agent first learns how to stand up and in the transfer task needs to learn how to walk.

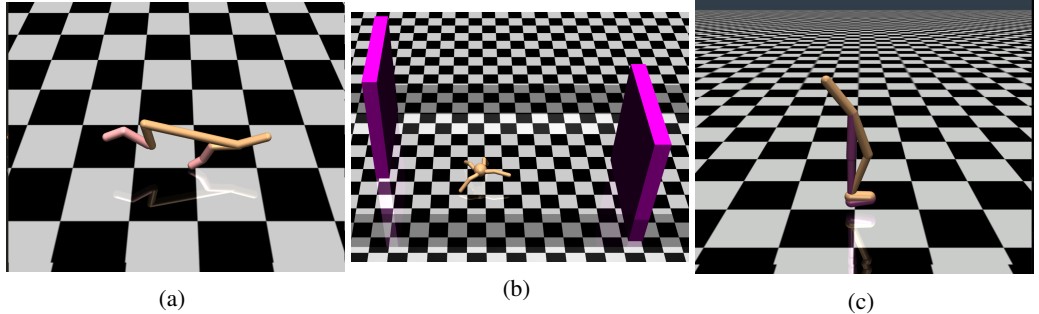

(a)  (b)  (c)

Figure 7: **MuJoCo** tasks visual representation.

Table 4: MuJoCo experiments hyperparameters

| Hyperparameter | AntWalls-v1 | Walker2dStand-v1 | HalfCheetahDir-v1 |
|---|---|---|---|
| PPO Learning rate | 2e-4 | 2e-4 | 3e-4 |
| IOC LR (policy, termination, master) | 1e-4 | 1e-4 | 1e-4,5e-4,5e-4 |
| MOC LR (policy, termination, master) | 8e-5 | 9e-5 | 3e-4,5e-4,5e-4 |
| IOPG LR (policy, termination, master) | 9e-5 | 8e-5 | 1e-4,3e-4,5e-4 |
| $\gamma$ | 0.99 | 0.99 | 0.99 |
| $\lambda$ | 0.95 | 0.95 | 0.95 |
| Entropy coefficient | 0.0 | 0.0 | 0.0 |
| LR schedule | constant | constant | constant |
| PPO steps | 2048 | 2048 | 2048 |
| PPO cliping value | 0.1 | 0.1 | 0.1 |
| # of minibatches | 32 | 32 | 32 |
| # of processes | 1 | 1 | 1 |
| $\eta$ | 0.9 | 0.9 | 0.9 |

### A.4.2 Implementation Details

We based our implementation on [Dhariwal et al., 2017, Klissarov et al., 2017, Khetarpal et al., 2020] and we ran the experiments for 2M steps for AntWalls-v1 and Walker2dStand-v1, and for 1M steps

for HalfCheetahDir-v1. After about one quarter of the total timesteps we change the original task to a transfer task that is meant to be more challenging.

For the Walker2dStand-v1 and AntWalls-v1 experiments, we performed a hyperparameter search for all algorithms on the learning rate in the set $\{8 \cdot 10^{-5}, 9 \cdot 10^{-5}, 2 \cdot 10^{-4}, 5 \cdot 10^{-4}, 7 \cdot 10^{-4}\}$. As in our previous experiments, we used the same learning for all options components. For the HalfCheetahDir-v1 experiment, we performed a hyperparameter search for all algorithms on the learning rate in the set $\{1 \cdot 10^{-4}, 3 \cdot 10^{-4}, 5 \cdot 10^{-4}, 7 \cdot 10^{-4}\}$. For this experiment we tried different values of the learning rates for each option component in order to stay consistent with the hyperparameter search done in previous work [Khetarpal et al., 2020]. For the MOC $\eta$ hyperparameter, we simply used $0.9$ as we have previously shown that higher values are usually better.

The architecture for the algorithms was kept to be the same with respect to the original code base. All other PPO hyperparameters were kept as is. All the values are available in Table 4.

### A.4.3   Option visualization

We share qualitative results of the kind of options our method, MOC, learns as well as the ones learned by the IOC approach. These results are available in the form of videos in the following URL.

### A.5   MiniWorld Navigation

We build on the first-person visual navigation domain of MiniWorld [Chevalier-Boisvert, 2018] to propose a transfer learning setting. In the transfer task we aim to increase the difficulty of the source. Moreover, we look at a setting where the agents could benefit from reusing the options learned in the source task. The tasks are defined as in the following.

We experiment with the **MiniWorld-YMazeTransfer-v0** shown in Figure 8a domain which contains of a maze in a Y shape. The agent spawns in a random position in the bottom part and has to reach the goal represented by a red box. At first this goal is in a random location on the left side and in the transfer task the goal is moved to the right side. We also explore the **MiniWorld-WallGapTransfer-v0** domain shown in 8b, where a gap in the wall separates two large rooms. At first, the agent has to navigate to the adjacent room, scan the environment and collect the goal represented by a red box. In the transfer task, we introduce a blue box, which now is the new goal and is located in the room adjacent to where the agent spawns. We also include a red box in the same room as the agent, however this box no longer gives the agent any reward.

For both environment, the agent and the goal locations are randomized at each iteration in order to avoid deterministic solutions.

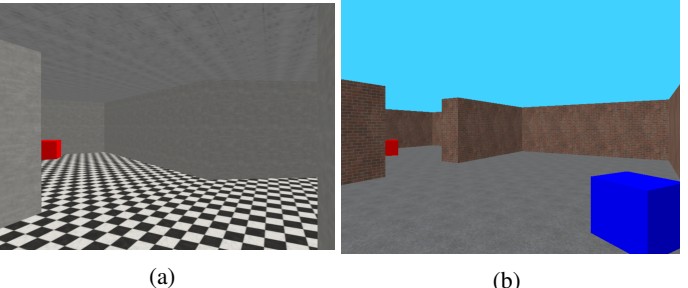

(a)                                                    (b)

Figure 8: **MiniWorld** tasks visual representation.

### A.5.1   Implementation Details

We based our implementation on [Kostrikov, 2018] and we ran the experiments for 2M steps for the MiniWorld-YMazeTransfer-v0 experiment and 5M steps for the MiniWorld-WallGapTransfer-v0 experiment. After 1M steps and 2M respectively, we change the original task to a transfer task that is meant to be more challenging.

The architecture for the algorithms was kept to be the same with respect to the original code base. Similarly, all PPO hyperparameters, including the learning rate, were kept as is. For the MOC

| Hyperparameter | Value |
|---|---|
| Learing rate | 2.5e-4 |
| $\gamma$ | 0.99 |
| $\lambda$ | 0.95 |
| Entropy coefficient | 0.01 |
| LR schedule | constant |
| PPO steps | 128 |
| PPO cliping value | 0.1 |
| # of minibatches | 4 |
| # of processes | 32 |
| $\eta$ | 0.7 |

Table 5: Hyperparameters for the MiniWorld experiments

algorithm we used $0.7$ as a value for the hyperparameter $\eta$ after looking for values in $\{0.7, 1.0\}$. Best hyperparameters values are found in Table 5.

### A.5.2 Option visualization

Once again we share results on the options learned by our MOC algorithm and the OC algorithm. These results are available in the form of videos in the following URL.

### A.6 Proof of Lemma 1

**Lemma 1.** *Under assumptions 1-2 the following limit exists,*

$$\lim_{t\to\infty} P_{\pi,\mu,\beta}(S_t = s, O_t = o|S_0, O_0) = d_{\pi,\mu,\beta}(s, o) \tag{2}$$

*and the augmented chain process is an ergodic Markov chain in the augmented state-option space. Moreover the stationary distribution can be decomposed in the following form,*

$$d_{\pi,\mu,\beta}(s, o) = \sum_{\bar{o}} \bar{d}_{\pi,\mu,\beta}(\bar{o}, s) p_{\mu,\beta}(o|s, \bar{o}) \tag{3}$$

*where $p_{\mu,\beta}(o|s, \bar{o})$ is the probability over options upon arriving in a state.*

*Proof.* The limit in Equation 2 exists if and only if the Markov chain on the augmented state-option space is ergodic. To ensure ergodicity, we first need to verify irreducibility, that is, whether it is possible to get from any state-option pair $(s, o)$ to any other pair $(s', o')$ in a finite amount of time. Concretely, there exist a positive integer $n$ such that,

$$[(P_{\pi,\mu,\beta})^n]_{(s,o),(s',o')} > 0$$

As we consider a finite augmented space, positive recurrrence is implied by irreducibility. Secondly, we need to verify that the Markov chain on the augmented space also satisfies aperiodicity.

To verify both criteria, we will use the following expansion on the state-option transition function,

$$P_{\pi,\mu,\beta}(s', o'|s, o) = p_{\mu,\beta}(o'|s', o) \sum_a P(s'|s, a)\pi(a|s, o) \tag{8}$$

$$= \big(\beta(s', o)\mu(o'|s') + (1 - \beta(s', o))\delta_{o,o'}\big) P^{\pi_o}(s'|s) \tag{9}$$

For each option, Assumption 1 verifies that $P^{\pi_o}(s'|s)$ is ergodic. Moreover, Assumption 2 implies that both the termination functions $\{\beta\}$ and the policy over options $\mu$ have strictly positive values. As per the definition in Equation 9, both assumptions lead to satisfying the irreducibility and aperiodicity criteria and therefore the chain on the augmented space is ergodic. We can then write,

$$\lim_{t\to\infty} P_{\pi,\mu,\beta}(S_t = s, O_t = o) = \sum_{\bar{o}} \lim_{t\to\infty} P_{\pi,\mu,\beta}(S_t = s, O_{t-1} = \bar{o})$$

$$\big((1 - \beta(s, \bar{o}))\delta_{o=\bar{o}} + \beta(s, \bar{o})\mu(o|s)\big)$$

$$= \sum_{\bar{o}} \bar{d}_{\pi,\mu,\beta}(\bar{o}, s)\big((1 - \beta(s, \bar{o}))\delta_{o=\bar{o}} + \beta(s, \bar{o})\mu(o|s)\big)$$

□