# OpenReview forum: "Flexible Option Learning"
_NeurIPS.cc/2021/Conference — NeurIPS 2021 Spotlight_

### Official Review · Reviewer_Q89w · 2021-07-02

**Rating:** 7
**Confidence:** 3

**Summary:**

This paper presents a method to augment Option-Critic based methods with a modified objective that allows all options to be updated with every sample, as opposed to just the option that was executed. Performance improvements over Option-Critic and other baselines on multiple environments, the flexibility of their objective to augment many similar methods, and evaluations that are mostly comprehensive are why I recommend this paper to be accepted.


**Limitations And Societal Impact:**

No explicit discussion on limitations, I think the authors should add this to the discussion section.


**Main Review:**

## Paper Strengths
**Motivation**
The motivation is intuitive and sensible. We want to be able to simultaneously update many options, instead of just one, so that we can encourage better sample efficiency and possibly more useful options.

**Mathematical Formulation**
The assumptions made to formulate the objective are sensible and the modified objectives are formulated in a principled manner starting from the one-step TD update.

**Method**
The method is a series of objectives that can be applied across many option-learning architectures. It is also, to my knowledge, novel.

**Experiments**
The authors test across 3 domains, with multiple environments in the continuous control domain. They also perform interesting analysis on Ant, with option visualizations that show the usefulness of the second option. There are modest to strong performance improvements across their environments over baselines.

## Paper Weaknesses
**Experiments**
One area where HRL/Option methods shine is sparse-reward, long-horizon tasks in continuous control (typically navigation environments). I would’ve liked to see evaluations on these types of tasks in the continuous control section. It’s also possible that visualizations on those environments would show interesting learned option behaviors.

## Questions
Shouldn’t Eq 1 be either $b_t + A_t \theta_t$ or there be $A$ be negated after line 117?

Why do you think the variance across runs is reduced despite any potential variance introduced by importance sampling?


**Time Spent Reviewing:**

6

---

> ### Author Response · Authors · 2021-08-10
> **Author Response**
>
> We are very glad that the reviewer finds our method sensible, novel and useful. Hopefully in the following we address adequately the reviewer’s concerns.
>
> **Experiments.**
> We agree with the reviewer and we would like to highlight that in section 5.3.2, the MiniWorld domain is a navigation-type task with sparse rewards and a relatively long horizon (especially for MiniWorld-WallGapTransfer-v0). For these experiments some interesting behaviour is discovered as shown in the videos linked in the appendix A.5.2. Perhaps a difference with continuous control is that these experiments are first-person point of view types, that is the algorithm receives images as input. If the reviewer still believes that more experiments in continuous control should be added we would be glad to try it out.
>
> **Equation 1.**
> We use the same notation as in (Mahmood et al., 2016, An Emphatic Approach to the Problem of Off-policy Temporal-Difference Learning, equation 4). We can’t seem to notice any discrepancies between the notation in that paper with what we present. That being said, in the literature sometimes this A matrix is written differently, as in (Bertsekas and Tsitsiklis, 1996, Neuro-Dynamic Programming, Section 6.3.3)
>
> **Reduced Variance.**
> A possible reason for the reduced variance is that our algorithm will perform updates to all options and weight these updates by its probability. In a sense this can be seen as an update in expectation. This is similar to the all-action updates for policy gradient methods (Sutton et al., 1983, Comparing policy-gradient algorithms), where all actions are updated in expectation. These updates are known to reduce the variance as they can be derived as conditional Monte Carlo estimators  (Bartley et al., 1987, A Guide to Simulation; Petit et al., 2019, All-action policy gradient methods: A numerical integration approach). A similar, yet slightly different phenomenon can be seen when considering true stochastic gradient descent (in the sense of taking a single datapoint for each update) versus mini-batch gradient descent.

---

### Official Review · Reviewer_KX8a · 2021-07-16

**Rating:** 6
**Confidence:** 4

**Summary:**

- This paper examines the possibility of improving sample efficiency in option learning by updating all options simultaneously, rather than the single option chosen for execution. The authors that posit that such an all-options update can improve sample efficiency both when a set of pre-computed options are provided to the agent a priori as well as when the agent must learn the options themselves from scratch along with how to best compose them
- The primary contributions of this paper center around the theoretical formulation that facilitates updating of multiple options derived from the experiences collected under a single option along with an empirical instantiation of the idea with tabular, linear, and non-linear function-approximation settings.
- Empirical results are presented confirming the efficacy of the proposed update when a set of pre-computed options are provided as well as when two options are being learned by the agent from scratch. Crucially, the authors are able to report modest performance gain across all tasks in the evaluation.

**Limitations And Societal Impact:**

I believe this is clear from my main review.

**Main Review:**

This paper scores highly on clarity, significance, and (to the best of my knowledge) originality. On that last point, these ideas seem analogous to those in the (flat) reinforcement learning setting [1] which may be an interesting point of exploration for more rigorous theoretical work in this space. However, I do see numerous issues on the axis of quality, with varying degrees of severity. If the authors can suitably resolve them, I would be happy to raise my score.

Based on how the authors have presented options, they have left themselves open to the more general setting where not all options are available at each state (as opposed to the fairly common, but less general, assumption that all options are available from all states). Under this setting, the initiation set does impact the value function definition and distributions outlined in Section 3. Fortunately, this can be fixed easily by simply defining the set of available options, as done for example by Khetarpal et al. 2020, or by introducing an appropriate initiation event and conditioning, as done in the original options paper by Sutton, Precup, & Singh 1999. That said, the rest of the paper seems to be assuming all options are available everywhere and, in fact, perhaps this is necessary to achieve ergodicity in Lemma 1 (otherwise some state-option pairs automatically become unreachable). In any case, the authors should either adjust the definitions accordingly or make this assumption explicit.

On the topic of Assumption 2, I wonder if there is some level of redundancy? In particular, I don't think one could achieve the ergodicity in Assumption 1 if either the termination function or the option policy yields a probability of zero for any state-option pair. Perhaps the authors should simply add the text of Assumption 2 to Assumption 1 so that the implications of ergodicity are made transparent to readers? That said, it seems like this assumption goes against the spirit of what options are trying to achieve. Naturally, we think of hierarchical multi-task reinforcement-learning agents endowed with a broad, diverse array of skills while the individual tasks presented to the agent may only demand a subset of them at a time. Even in the single-task setting, we expect some skills to have a natural sequentiality such that one would never execute skill B before having reached a particular state from the termination of skill A. Could the authors comment on why Assumption 2 is reasonable given the scale at which we expect hierarchical reinforcement-learning agents to be able to operate? Otherwise, it seems like the long-term possibilities for the proposed algorithm are quite limited.

In Lemma 1, I would have expected the stationary distributions appearing on the LHS and RHS of Equation 2 to match. Instead, the authors introduce a new distribution on the RHS, \bar{d}, which hasn't been defined anywhere in the paper. If this is not a typo, could the authors please clarify what this distribution represents? In the rest of this review, I'll assume this was a typo and the two distributions were meant to be the same, evaluated with different arguments on the LHS and RHS.

The authors correctly introduce new notation in L151 to distinguish the discounted stationary state-option distribution from the distribution defined by Equation 2. Given the analogous discussion for standard policy-gradient methods in (flat) reinforcement learning [2], could the authors comment on the mismatch between these two distributions, with an algorithm only being able to generate samples from the latter, and how that might affect the use of the intra-option policy gradient in their experiments (I suspect the effect would be shared among all the OptionCritic methods evaluated, not just MOC)? This question, while interesting, is perhaps beyond the scope of this work and, accordingly, the authors should not prioritize answering it in their rebuttal if space is constrained.

In Section 4.4, the introduction of importance-sampling ratios introduces two new assumptions of absolute continuity (one for each ratio used in the equation of that section). From a theoretical perspective, this again comes back to a question of reasonability for these assumptions and why it should be desirable for specific option policies to necessarily retain non-zero probability mass on all actions so as to entertain these off-policy updates? Intuitively, these assumptions seem overly restrictive and unreasonable. From a practical perspective, I wonder if there are any unmentioned reguarlization techniques or other heuristics (for example, entropy regularization) used to ensure that these absolute continuity conditions were met in the experiments?

The information radius criterion used in Figure 1c seems rather opaque as it is currently discussed. It would be of great benefit to readers if the authors could offer a bit more insight into the measure itself to clarify the importance of the result.

In Algorithm 1, the upon-arrival probability (roughly line 14) is currently an unbound variable; the authors should include psuedocode for its computation. The same is true of V_\Omega(s') two lines later although, presumably, this is just taking the expectation on Q_\Omega with respect to \mu_z.

One missing hyperparameter I can't find explicitly stated in each experiment is the total number of options learned. From scanning through, it seems as though no experiment learns more than two options; could the authors please confirm this? A natural follow-up question here is if the authors experimented with more options? Otherwise, it seems a bit difficult to claim that the proposed method does not suffer from issues of diversity when learning useful options (as diversity of two things is a rather low bar to clear). For a domain like FourRooms, I would have wanted one experiment to use precisely four options just to test that MOC does not fall into the degenerate solution of aliasing the four actions of the original MDP. I also wonder if this limitation to two options is done so as to allow the experimental setup to adhere to the aforementioned assumptions of this work; does MOC begin to break down when trying to learn 10 options? 20 options? 30?

===== Post-Rebuttal =====

I thank the authors for their response and clarifying many of the points I raised. I'm optimistic that they can suitably adjust the text in the main paper to reflect these clarifications.

I still find the need to keep non-zero probability mass on all actions/options for all policies to be restrictive. Even in a transfer setting, the margin for uncertainty mentioned in the author response could simply pertain to a decision of whether or not the agent should attempt to expand its library of options/skills in this task as opposed to discerning optimal behavior solely with the options on hand.

More importantly, I believe that further empirical investigation is needed to investigate MOC when learning far more than 2 options, and on domains far more complex than MountainCar. The now classic OptionCritic paper seems to go as far as 8 and, genuinely, if the premise of MOC to allow for updates across all options holds, then we should be able to push far past that point or, in an equally-important fashion, shine a spotlight on precisely where the breaking point resides. Even if the authors were to run such additional experiments (which I am not asking them to do), the critical piece would be the result and associated exposition in the main paper, which currently cannot be judged in the post-rebuttal phase. I'm incrementing my score to reflect the clarifications made, however, I believe the authors would be best served in providing a more rigorous empirical study with more learned options to sharpen their contribution. Hopefully, if the paper is accepted here, the authors will consider doing this anyways for the camera-ready version as I suspect this question/point of concern is a natural one apparent to anyone in the community hoping to leverage this approach.

[1] Ciosek, Kamil, and Shimon Whiteson. "Expected Policy Gradients for Reinforcement Learning." Journal of Machine Learning Research 21 (2020): 1-51.

[2] Nota, Chris, and Philip S. Thomas. "Is the Policy Gradient a Gradient?." In Proceedings of the 19th International Conference on Autonomous Agents and MultiAgent Systems, pp. 939-947. 2020.

**Time Spent Reviewing:**

3

---

> ### Author Response · Authors · 2021-08-10
> **Author Response (Part 1)**
>
>
> We would like to thank the reviewer for their dedication. We are very glad to see that the reviewer appreciates the significance and originality of our work. We hope that the following answers will contribute to improving the quality and clarity of the paper. Also we will include Ciosek et al. 2020 in the related work as more connections could be drawn by future work.
>
> **Initiation sets.** This is a good point and we will make sure that it is clearly presented. In particular, we explicitly do not mention the concept of initiation sets in the paper as to not encumber the notation and equations. As the reviewer rightfully mentions, by considering the concept of interest functions from Khetarpal et al. 2020, it would be possible to extend these definitions to the general case. To address this issue, we can either add a section in the appendix where the interest function is included in the definitions or directly change the notation in the paper.
> With regards to Lemma 1 and initiation sets, there is indeed a nuance to be added in the paper. If we consider the original concept of initiation sets, then we might not be able to use the Lemma as-is since some state-options pairs are unreachable (as initiation sets map states to {0,1}). However, if we consider the more general concept of interest functions, our results apply automatically as these functions map states and options to the positive real numbers. From a practical point of view, utilizing interest functions can also be advantageous as they are parametrized functions that can be learned by current RL methods.
>
> **Assumptions.** We will make sure to clarify why each assumption is important in the paper. The importance of assumption 1 is to consider how the option policies behave on their own. In other words, each option policy would need to induce an ergodic Markov chain over the whole state space (notice that this is not the state-option space as in Lemma 1), irrespective of how the termination function behaves. By looking at the behaviour of option policies we do not consider its interaction with the termination function or other option policies. We then introduce assumption 2 in order to precise how this interaction with other quantities should be defined (and how it behaves over the state-option space).
>
> It is true that HRL is about learning options/skills and knowing when to use a subset of them, and we fully agree on this motivation. A way to reconcile this concept with our method is once again to consider interest functions, which are compatible with our contribution, instead of binary initiation sets. On one hand, the interest functions are learnable, which scales with long term possibilities of HRL. On the other hand, the concept of interest functions provides a way to strongly disfavour a subset of options by assigning an infinitesimal value, such as $\epsilon$, to some options and favour some other options with high values such as $1 - \epsilon$. As the reviewer mentions, perhaps in a single task this $\epsilon$ might not be necessary. However, it can be argued that in a transfer or multi-task setting (which we focus on), it can be advantageous to use $\epsilon$ in order to have a margin of uncertainty on whether an option applies to state on an unseen task.
>
> Finally, we also want to point out that the assumptions only apply for the option evaluation case, and as such the updates we propose for control (Eq 6) could still be applied even if some state-option pairs are unreachable. We will make this clear in a revised version of the paper.
>
> **Lemma 1.** We should have been more precise in the way we introduce this notation, especially in the appendix. The $\bar{d}$ notation is to specify the limiting distribution over state and previous option pairs. That is, $\bar{d}(\bar{o},s) = \lim_{t \rightarrow \infty} P_{\pi,\mu,\beta}(O_{t-1}=\bar{o},S_t=s)$. We notice now that a typo in the last line of the proof of the Lemma might have contributed to the confusion, which we will correct. We want to emphasize that this distribution is not exactly the distribution over state and option pairs $d(s,o)$, as arriving to a state with an option has a different probability then having selected an option in a state.
>
> **Policy Gradient Methods.** We appreciate the reviewer noticing the change in notation, which is sometimes ignored in papers. The reason why in practice PG methods do not scale the updates by $\gamma^t$ is still poorly understood. We have some very interesting results in Nota et al. 2020, as mentioned by the reviewer, which indicate that currently our PG implementations are not the gradient of any objective. These results are just as valid in the hierarchical setting, for any methods based on PG updates, including the original Option-Critic framework. There is some recent work (Zhang et al. 2020, A Deeper Look at Discounting Mismatch in Actor-Critic Algorithms) which takes a representation learning point of view to explain the current PG methods. However, it’s still not clear why such a scaling is not applied. It is possible that at the moment we rely too much on the discounted setting, in theory as well as in practice, and that we should consider moving towards the average reward setting. We suspect that our update rules would extend naturally in that setting as well.

---

> > ### Author Response · Authors · 2021-08-10
> > **Author Reponse (Part 2)**
> >
> >
> > **Importance Sampling.**
> > We would first like to mention that no additional heuristics have been implemented for our methods.  In the experiments on MiniWorld, we followed the standard practice of adding entropy regularization to the actor updates when the action distribution is discrete (we haven’t changed the value of this hyperparameter). In deep RL, this regularization is done even in the on-policy setting and seems to be done to improve performance. In our experiments, this standard regularization is applied to all the methods (including the PPO agent). Additionally, this regularization is not added to the MuJoCo experiments as it is not standard practice in that continuous control.
> >
> > The absolute continuity assumption that comes from using importance sampling ratios seems to be prevalent in off-policy methods that convergence under linear function approximation and only require single time-step updates, rather than two time-step such as Gradient Temporal Differences (Maei et al., 2011) or ABQ (Mahmood et al., 2017). It’s not clear yet how we can avoid such an assumption. It could be possible to consider the Deterministic Policy Gradient and use DDPG in practice, but it should be noted that Equation 15 from (Silver et al., 2014, Deterministic Policy Gradient Algorithms) relies on the same approximation made in (Degris et al., 2014, Linear off-policy actor-critic). This approximation is subsequently addressed in the errata in Degris et al. 2014, which mentions that the update rule is then only convergent for tabular representations. When we move to function approximation we would need to employ updates following (Imani et al., 2018,  An Off-policy Policy Gradient Theorem Using Emphatic Weightings), which introduces IS weights as well.
> >
> > We would also like to motivate the idea of having non-zero values for actions/options (that is stochastic policies instead of deterministic ones) as being perhaps more appropriate for the transfer/continual learning setting. This setting is especially appealing for hierarchical methods as we aim to transfer skills/options across many tasks. By providing a natural way to incorporate uncertainty in the policies, we believe it might be advantageous and lead to better adaptation.
> >
> > **Information Radius.**
> > We use the notion of information radius as it is a way to generalize the KL divergence beyond two distributions (Sibson et al. 1969, Information Radius). Formally it is defined as $IR = (1/n) \sum_{i}^n [ KL(P_i | \sum_j^n P_j (1/n) )]$ where $n$ is the number of options. We will add this definition to the revised version of the paper.
> >
> > **Algorithm 1.**
> > We believe there might have been a misunderstanding, and we will clarify this in the paper. The variable U is a sample of the discounted return. It is taken from Chapter 9 of Sutton and Barto 2018, but perhaps we should use a different letter to avoid confusion with the upon-arrival distribution. We will make sure to define it clearly, as well as $V_\Omega$ and $Q_\Omega$.
> >
> > **Number of options.**
> > For the experiment on FourRooms with fixed options we used 12 options (classically defined as the “hallway” options), whereas the experiments with learned options we tried using 2 options and 4 options. Figure 1b shows the results for 4 options. In the appendix A.2 we plot the kind of options that are learned for the case of 2 options as we believed it might be easier for a reviewer to understand the results at a first glance. We will add the qualitative results for 4 options in the revised version.
> >
> > For the MountainCar experiments, we only used 2 options. However, we ran additional experiments to address the reviewer’s point and we report the results for the case of 4 and 8 options in this [link](https://drive.google.com/drive/folders/1T02BeRF9YKSHiUzSVKpCpeeu8fzauZTA?usp=sharing). In terms of performance, it seems like 4 options seems to work better for this domain, especially when looking at the figure before the transfer happens. In terms of the information radius, it also seems like 4 options provides a more diverse set of option policies. It would be interesting to evaluate further if this correlation exists in other domains. However, we note that for the case of 8 options there seems to be a slight drop in performance and diversity.  We can try running additional experiments with even more options if the reviewer is interested.
> >
> > For the experiments at scale (deep RL) we used 2 options in order to allow a natural comparison with previous work that also only uses 2 options. Another reason is that these experiments intuitively can be divided into two distinct policies. We believe that interpretability is an important feature of hierarchical methods, and as such using two options brings more clarity to these experiments. It is likely that using many more options might reduce interpretability, even if it can potentially increase the final performance (this increase in performance could come from better representations or by having considerably more parameters). We believe that HRL research has a lot of potential but currently it is also in its early phase, and we therefore chose to proceed cautiously. We can add these points as a possible limitation to our method to a revised version of the paper. We could also perform experiments with many more options if the reviewer deems this to be necessary.

---

### Official Review · Reviewer_XUis · 2021-07-20

**Rating:** 8
**Confidence:** 4

**Summary:**

The paper proposes a principled way to update all options at once in an option-learning framework, and conducts extensive experiments to support the proposed algorithm.

**Limitations And Societal Impact:**

The paper did not seem to address limitations at all. It would be good to state that using importance sampling rules out deterministic policies, clarify whether assumption 2 rules out the use of initiation sets, and state any further limitations that the authors are aware of.

**Main Review:**

I recommend accepting this paper for publication, conditional on the issues listed below being addressed in the final version.

**Quality:**

Overall the quality is very good. The proposed method is principled, and the experiments are numerous, extensive, and well-done.

**Clarity:**

The paper is written very clearly and is easy to understand. There are some grammatical errors throughout (the ones I noticed are listed below).

**Significance:**

The proposed method is very general and very effective, making it likely to be significant moving forward.

**Originality:**

The proposed algorithm is novel to the best of my knowledge.

**Misc. comments, questions, and suggestions for improvement:**

Line 89: should be "selected"

Line 99: "This options" should be "This option"

It's good to comment on the assumptions made in the paper, explaining why they're made, whether they're common, how restictive they are, etc.

Could the authors justify Assumption 2, and briefly comment on it in the paper? Assumption 2 seems to conflict with the idea of an initiation set by implying that the initiation set for all options must be all states. Presumably the initiation set was included in the options framework for a reason, so it would be good to comment on why this assumption is being made. Is it a simplifying assumption analogous to assuming that all actions are available in all states?

Line 130: "each" is repeated.

The discount factor can be interpreted as the probability of the agent terminating at each timestep. So why is the paper using discounting in addition to the termination function of each option?

Line 177: "hyperparamter" should be "hyperparameter". This also occurs in the Figure 2 caption.

Line 191: "that" should be "than"

Line 197: Missing a "to"

Line 205: "fo" should be "of"

Section 4.4: Using importance sampling in this way can lead to high-variance updates, right? Is this variance issue somewhat mitigated by the hyperparameter $\eta$ introduced in section 4.3? Additionally, using importance sampling rules out deterministic policies, which is a limitation of the algorithm that wasn't mentioned in the paper. In fact, the appendix describes how the original FourRooms options had to be modified to be stochastic so that they would work with importance sampling. It would be better to be clear about how exactly they were modified, and how incorporating importance sampling limits the policies allowed.

Line 234: "a good separation of the state-state has been learned."

Figures: What are the shaded regions? Confidence intervals? Standard error? It's best to plot confidence intervals because non-overlapping confidence intervals imply statistically significant results, whereas non-overlapping standard error does not. That way the reader can see at a glance whether results are statistically significant or not. If the confidence intervals end up overlapping, it can be ok to lower the confidence level; it still provides some evidence to support the proposed method.

Experiments: When a paper uses "ours" ("our approach", "our method", etc.) excessively it sounds like an annoying advertisement. Framing experiments this way makes it feel like the authors' only goal is to showcase the strengths of the proposed method, instead of conducting a fair and impartial comparison. What's weird is that the experiments seem to be done pretty fairly, so why the excessive salesmanship? It was starting to really annoy me by the end of the paper.

Relatedly, the discussion of some of the figures (Figure 2a, Figure 5a for example) feels biased and bordering on overclaiming. The results are really good; there is no need to exaggerate (discussion of figure 2a), and it's ok to point out situations where MOC took longer to learn initially (figure 5a)---it could be an avenue for further research.

Figure 2: What value of $\eta$ was used in 2a?

Line 259: should be "number of updates"

Appendix A.3.1: The caption for Table 2 says "FourRooms" when it should be sparse mountain car.

I was initially wary of the introduction of the hyperparameter $\eta$, because often papers will introduce a new hyperparameter, search for the best-performing value, and then claim a performance improvement with no guidance on how to set the hyperparameter on a new environment without testing. However, I was very happy to see that $\eta=1$ was (close to) the best performing value for the function approximation setting across different forms of function approximation and different base algorithms, which is really cool!

Line 327: "simultaenously"

**Time Spent Reviewing:**

10

---

> ### Author Response · Authors · 2021-08-10
> **Author Response**
>
>
> We would like to thank the reviewer’s effort for a very detailed and thorough review. We will make sure to address all the grammatical errors and typos (thank you for not penalizing these easily fixed mistakes!).  We try to answer the reviewer’s points in the following, please do let us know if other concerns appear.
>
> **Discussion around assumptions.**
> This is a good point and we will make sure to expand on it in the revised version of the paper.
>
> Regarding Assumption 1, it is usually done in the “flat” reinforcement learning setting in order to derive the stability and convergence of temporal-difference algorithms. These derivations rely on the existence of the limiting distribution, which itself depends on the ergodicity of the induced Markov chain. In the “flat” reinforcement learning setting, a single policy is assumed to induce an ergodic chain, whereas in the hierarchical setting we need to extend this assumption to all option policies as each of them may generate samples used for learning.
>
> For  Assumption 2, the reviewer brings another good point that we will make sure to highlight. It can indeed be seen as assuming that all actions are available for every state. This assumption is necessary for the general setting (that is without any more knowledge on the MDP) as we need to be able to ensure that any state-option pair can be visited with probability $\epsilon$, which can be very small, but must be greater than 0. With regards to initiation sets, a natural way to address this limitation is to consider the concept of interest functions from (Khetarpal et al. 2020,Options of Interest: Temporal Abstraction with Interest Functions), which is a generalization of the concept of initiation sets. In Sutton et al. 1999 an option’s initiation set maps a state to {0,1}, whereas an option’s interest function maps a state to the positive real numbers. As a consequence of this generalization, it is possible to learn the interest function by gradient descent, which is useful in practice.
> Related to our work, this generalization of initiation sets would then ensure that any state-option pair can be visited, even though this probability might be infinitesimal. We will make sure to include this discussion on initiation sets/interest functions and extend Assumption 2 to include the possibility of using interest functions.
>
> **Discount factor.**
>  It is true that the discount factor can be interpreted as the probability of the agent terminating, which is closely related to the termination function (indicating to the agent if the option should be stopped). One difference is that in practice, the discount factor is usually not sampled (it is simply included in the return), whereas the termination is sampled at every timestep. This illustrates the fact that the discount can also be seen as somewhat of a “mathematical convenience” (to cite Harutyunyan et al. 2019, Per-decision option discounting), in the sense that it makes things easier theoretically. It also seems to be necessary in practice for current RL algorithms to converge in experiments. That being said, this close connection between the discount factor and termination function is investigated in Harutyunyan et al. 2019, where the authors propose the concept of time dilation which encourages reasoning at the level of options. This seems like an interesting path to explore further and to obtain empirical evidence as to its practical advantage. We will include a discussion on this in the revised version.
>
> **Importance sampling (IS).**
>  Using IS can lead to high variance, which in practice can lead to divergence. The hyperparameter $\eta$ introduced in Section 4.3 is not exactly introduced to tackle this issue, as in our experiments its value is either 1 or close to 1. We introduced $\eta$ in order to verify how updating all options might affect the diversity of the option set, as well as provide a more general method. However, the variance of IS ratios could be countered by the fact that our suggested update rule is all-option update, or in other words it is an update done in expectation. This can lead to reduced variance, which could possibly counteract the variance introduced by IS. It is true that our method requires the option policies to be deterministic and we didn’t mention it in the paper. We will definitely make this clear as it might be an area of future investigation.
>
> **Shaded regions.**
>  For all experiments, except the MiniWorld ones where we plotted the standard error, we used a confidence interval of 80%. This discrepancy comes from using different code bases. We have since plotted the 80% confidence interval for the same runs on MiniWorld and we make these plots available in this [link](https://drive.google.com/drive/folders/14qD4nocPNlRQeXxsgA8syRR3SBKucJi6?usp=sharing).
>
> **Experiments and figure discussion.**
>  We agree with the reviewer’s point and the language around the discussion on the experiments can definitely be nuanced. We appreciate the reviewer’s feedback on this aspect as we believe it can improve the paper by encouraging a broader discussion. Adding to the example the reviewer points out (figure 2a and 5a), we could also mention that in Figure 3, the shaded area around the option duration plot is still quite large, which could indicate that we can still improve this aspect for hierarchical methods. Another aspect of the results that we refer to in the paper (L299) is that interpretable options are not learned for all random seeds. We will emphasize this by providing some visual examples of such runs.
>
> **Hyperparameter.**
>  Thank you for your motivating comment!  We thought that our approach could potentially lead to lower diversity and that this would be a limitation brought up during the reviewing process. This still seems to be the case for tabular experiments.

---

### Decision · Program_Chairs · 2021-09-27

**Decision:**

Accept (Spotlight)

**Comment:**

This paper presents an intuitive approach to updating multiple options together while learning in a hierarchical deep reinforcement learning setting. The approach seems principled, and performs well empirically.
All reviewers have advocated for acceptance, with the expectation that the requested clarifications in the paper are made.